# Chemical Composition of the Essential Oil of the New Cultivars of *Lavandula angustifolia* Mill. Bred in Ukraine

**DOI:** 10.3390/molecules26185681

**Published:** 2021-09-18

**Authors:** Katarzyna Pokajewicz, Marietta Białoń, Liudmyla Svydenko, Roman Fedin, Nataliia Hudz

**Affiliations:** 1Department of Analytical Chemistry, University of Opole, 45-052 Opole, Poland; katarzyna.pokajewicz@uni.opole.pl (K.P.); marietta.bialon@uni.opole.pl (M.B.); 2Sector of Mobilization and Conservation of Plant Resources of the Rice Institute of the National Academy of Agrarian Sciences, Plodove, Kherson Region, 74992 Kherson, Ukraine; svid65@ukr.net; 3Department of Pharmacy and Biology, S. Z. Gzhytsky Lviv National University of Veterinary Medicine and Biotechnology, 79010 Lviv, Ukraine; roman_fedin@ukr.net; 4Department of Drug Technology and Biopharmaceutics, Danylo Halytsky Lviv National Medical University, 79010 Lviv, Ukraine; 5Department of Pharmacy and Ecological Chemistry, University of Opole, 45-052 Opole, Poland

**Keywords:** *Lavandula angustifolia*, lavender, essential oil, GC-MS, cultivar, growth year

## Abstract

Lavender, otherwise known as *Lavandula angustifolia* Mill., is widely used in landscaping, and its oil is a valuable raw material used in many industries. Therefore, new varieties of this plant are bred. The essential oil composition obtained from fresh flowers of thirteen new Ukrainian cultivars of *L. angustifolia* were analysed by GC-MS, and eighty-two components were identified. Linalool and linalyl acetate were principal constituents of all of the samples, and ranged from 11.4% to 46.7% and 7.4% to 44.2%, respectively. None of the studied samples fulfilled the requirements of Ph. Eur. and ISO 3515:2002. The main reason was a high content of *α*-terpineol (0.5–4.5%) and/or terpinene-4-ol (1.2–18.7%). Our results are in line with multiple researchers showing that the studied lavender oils do not comply with the industry standards despite their authenticity. We also investigated the effect of the growth year on the chemical composition of five tested cultivars grown on the same plots and noticed a considerable variability between years. The obtained experimental data did not show a significant inter-year trend for the content changes of the major components. Our results allow us to deeply characterize the new cultivars and evaluate their oil for a possible use in the industry, or to designate them for future selective breeding.

## 1. Introduction

The Lamiaceae family contains many aromatic and medicinal plants [1,2,3]. One such plant is *Lavandula* genus, which embraces valuable herbs and covers about thirty species, dozens of subspecies, hundreds of hybrids, and selected cultivars [4,5,6,7,8,9]. Four species of this genus are widely used in the cosmetic, perfume, and pharmaceutical industries, namely (1) *Lavandula angustifolia* Mill., commonly known as English lavender (formerly known as *L. vera* or *L. officinalis* or true lavender); (2) *Lavandula stoechas* L., which has a very strong odour, sometimes known as French lavender; (3) *Lavandula latifolia* Medik., a Mediterranean grass-like lavender; and (4) *Lavandula x intermedia* Emeric ex Loisel., which is a sterile cross between *L. latifolia* and *L. angustifolia* [4,5,6,10]. They are extensively cultivated in some countries, especially Bulgaria, China, France, Australia, Morocco, Spain, Ukraine, and the United Kingdom [4,5,9]. Among them, *L. angustifolia* is considered to be the most important species of this genus. Its essential oil is highly valued due to its attractive fragrance, low camphor content, and the fact that the oil yields are less than the yields of spike oil (from *L. latifolia*) or lavandin oil (from *L. x intermedia*). Therefore, the oil of *L. angustifolia* is much more expensive and it is often adulterated with the cheaper above-mentioned oils [1,5,6,7,8,11,12,13]. It is also referred to as true lavender oil, and it has been used for cosmetic and medicinal purposes for centuries. Nowadays, it is extensively used in the perfume and cosmetic industries as it contains more than one hundred components, including linalyl acetate, linalool, terpinen-4-ol, lavandulol, lavandulyl acetate, 1,8-cineole, limonene, *cis*- and *trans*-*β*-ocimene, etc. [5,7,8,11,12,13,14]. Moreover, 60–90% of all cosmetic products contain linalool, which has a floral, citric, fresh, and sweet odour [8]. Lavender oil is used for flavouring beverages, ice cream, candies, chewing gums, and baked goods in the food industry [14]. 

The medical administration of true lavender oil is still practised [6]. The oil is approved as an herbal medicine by the European Medicine Agency [15]. Most commonly, lavender oil is recommended for oral administration. Several animal and human studies suggest that it has anxiolytic, mood stabilizing, sedative, anti-inflammatory, analgesic, anticonvulsive, and neuroprotective activities [6,15]. In addition, *L. angustifolia* oil is active against many species of microorganisms [11]. The lavender oils from fresh and dried aerial parts and flowers of *L. angustifolia* from Wielkopolska (Poland) demonstrated high activity against bacteria (*Bacillus subtilis, Staphylococcus aureus, Escherichia coli, Pseudomonas aeruginosa*), yeast, and filamentous fungi (*Candida* sp., * Aspergillus niger*, and *Penicillium expansum*), inhibiting their growth at the concentrations in the range of 0.4 to 4.5 μg/mL [8]. Furthermore, lavender oil and its major components, linalool and linalyl acetate, are used in aromatherapy by inhalation and aromatherapy massage [5,6,15,16]. Lavender inhalation may have a temporary effect on heart rate due to parasympathetic modulation. Aromatherapy was carried out by inhalation for 20 min after instilling 0.25 mL of the essential lavender oil (Australian Certified Organic Pty Ltd., Brisbane, Australia) and 50 mL of water into an ultrasonic ionizer aromatherapy diffuser. Midlife women with insomnia received a 12-week aromatherapy course twice a week (24 times total). These women experienced a significant improvement in sleep quality; however, lavender aromatherapy did not confer a benefit on heart rate variability in the long-term follow-up. Unfortunately, the chemical composition of the used essential oil was not indicated [16]. Other studies found that the inhalation of lavender oil reduced depressive moods and the systolic and diastolic pressures upon short-term exposure to lavender due to its relaxing effects [15]; however, the chemical composition of the used essential oil was not studied as thoroughly.

Lavender oil is also studied regarding its anti-inflammatory properties. The essential oil was effective against TPA-induced inflammation in mouse ears and showed better anti-inflammatory activity compared to ibuprofen at the same dosage. The inhibition effect at a dose of 100 mg/kg on TPA-induced mouse ear oedema was about 58.7%. The expression levels of the transcription factor, the nuclear factor kappa-B, inflammatory cytokine, cyclooxygenase-2, and the tumour necrosis factor were also decreased significantly. The principal identified components of the tested lavender oil were linalyl acetate (28.9%), linalool (24.3%), caryophyllene (7.9%), trans-3,7-dimethylocta-1,3,6-triene (4.6%), 4-terpineol (4.0%), lavandulyl acetate (3.5%), borneol (2.60%), and eucalyptol (2.05%) [17]. Cardia et al. (2018) also revealed that the lavender essential oil had anti-inflammatory activity. The treatment at the doses of 75 and 100 mg/kg significantly reduced the myeloperoxidase activity by 57.4% and 62%, respectively, similar to the activity observed with promethazine, as a reference drug product, at a dose of 10 mg/kg (65.1% of reduction) in male Swiss mice (weighing 20–30 g); however, the essential oil at the doses of 50 and 250 mg/kg did not significantly decrease the myeloperoxidase activity. The mechanism of anti-inflammatory activity involves the participation of prostanoids, nitric oxide (II), proinflammatory cytokines, and histamine [18].

Regarding chemical composition, lavender essential oil is characterized by a high content of linalool and linalyl acetate, a moderate amount of terpinene-4-ol, lavandulyl acetate, and lavandulol, and variable levels of eucalyptol (1,8-cineol) and camphor [1,5,11,12,13,19,20,21]. These are only a few components out of over 100, which contribute to the physicochemical and biological properties of the oil. The chemical composition of the essential oil of *L. angustifolia* can be highly varied, and is mainly determined by a plant genotype [1,19]; however, environmental factors, ontogenetic factors, region, conditions of cultivation, post-harvest processing procedures, and a part of the plant also influence the chemical compositions of the essential oil [1,8]. Nevertheless, the chemical composition of the *L. angustifolia* essential oil is regulated by the European Pharmacopeia (Ph. Eur.) and some international and local standards [13,20,21]. The structures of the most reported and regulated components of the essential oils of *L. angustifolia* are provided in Figure 1.

The primary aim of our studies was to study the essential oil composition of the new cultivars of *L. angustifolia* and discuss results in regard to literature and current industry standards. The secondary aim was to investigate inter-year variability in the oil composition. The obtained results are a part of the broader characteristics of the new lavender cultivars and will help in their evaluation concerning possible essential oil use in the industry and the selection of cultivars for further breeding.

## 2. Results and Discussion

The Rice Institute of the National Academy of Agrarian Sciences of Ukraine is dealing with the selection of new cultivar plants, including aromatic plants with high yields of essential oil, good decorative qualities, and frost and drought resistant properties for a further possible usage in the pharmaceutical, food, cosmetic, and perfume industries. Among such plants is *L. angustifolia*. Thirteen new cultivars bred in the Institute are discussed in this paper. They are listed and shortly characterized in the Section 3. Some of the cultivars are already cropped in Ukraine, and some are only used for improving selective works. Various lavenders have similar ethnobotanical properties and major chemical constituents. The main constituents of lavender are linalool, linalyl acetate, 1,8-cineole, *β*-ocimene, terpinen-4-ol, and camphor. The lavender essential oils are characterized by high levels of linalool and linalyl acetate [1,5,7,8,11,12,13,19,20,21]; however, the relative level of these constituents varies in different species, and even can be changed within one cultivar depending on the year of growing [1,8,22]. Therefore, we evaluated these new cultivars regarding their chemical composition by GC-MS.

Eighty-three different oil components (Table 1 and Table 2) were identified (or tentatively identified). It is worth noting that we experienced some difficulties with the identification basing on the use of a mass spectra library search only. These difficulties are already well described in the publication by Shellie et al. (2002) [19]. The correct identification of many compounds without an additional tool like Van den Dool and Kratz’s linear retention indices (LRIs) would be impossible. Therefore, all of the MS identifications were confronted with the indices from the library and literature data [23,24]. In one case, to prove some persistent MS peak misidentifications, analytical standards were needed. All of the taken measures allowed us to identify most of the detected peaks. Some minor components of the studied lavender oil samples could not be identified. Most of the likely inaccurate spectra were obtained due to the fact that library search did not find the hit and a manual visual mass spectra comparison with the terpenes reported in lavender oil did not allow to identify some detected components. Moreover, a comparison with terpenes from the library with close LRI values did not help [23]. The examples of those unidentified components are components 67–69 with LRI 1338, 1344, and 1354. As these samples of the essential oils were obtained from the new cultivars, it was impossible to exclude the possibility of some new chemical occurrences. These components were especially abundant in the oil from cultivar 4 (2018)—about 6%. The example of the total ion chromatogram (TIC) is provided in Figure 2.

Ten major components of all of the tested samples are presented in Table 3. These compounds accounted, usually, for more than 80% of the total chemical composition. The main components of all of the studied lavender oil samples were linalool and linalyl acetate (Figure 3). Their domination was evident as expected in lavender oil, and they made up 11–47% and 7–44% with a median of 36% and 30% for linalool and linalyl acetate, respectively. It is worth mentioning that the linalyl acetate content for cultivar 13 was much lower than for other cultivars for both years, with an outlying value of about 7%. Most probably, this is a characteristic feature of this cultivar. The third most abundant component was terpinen-4-ol with a largely variable content of 1–19% and a median of 7%. It seems that the terpinen-4-ol content for most cultivars (1,3,7–10, 12 and 13) was higher than the values usually reported. A literature review indicates values from 0.11% to 8%, with most results laying around 3% and rarely exceeding 6% [1,7,11,19]. The other main components were significantly less abundant and mostly far below 5%, and they included (in order of descending quantities): lavandulyl acetate, α-terpineol, *cis*-linalool oxide, geranyl acetate, *trans*-linalool oxide, neryl acetate, and borneol. Lavandulol and lavandulyl acetate are considered marker compounds for lavender essential oil [5]. In our samples, the lavandulyl acetate quantities were between 2.7% and 6.0%, except for cultivar 9 with a high content (12%), and cultivars 2 and 11 with a very low content (0.2–0.4%). Regarding lavandulol, we could distinguish two subgroups of the cultivars: lower contents (≤ 1%; cultivars 2–8, 10–12) and higher contents (≥2.6%; cultivars 1, 9 and 13). It is necessary to note that cultivar 2 did not meet the requirements of the Ph. Eur. for the lavandulol content (min. 0.1%).

The quality of lavender oil is an ambiguous issue. Even oil manufacturers and distributors have problems with the clear determination of the oil quality. One certain thing is that the adulteration of true lavender oil is not welcome, and the adulterated oil is recognized as low-quality oil. Furthermore, some lavender oil origins are more valued than others [5,12,20]. Two factors are important for the lavender oil quality: a pleasant aroma (a highly subjective feature) and a desired composition of components. The latter feature may be measured, and lavender oil composition is regulated by many international standards, including the International Standard Organisation (ISO), GOST (Russian Technical Standards), and Ph. Eur. [13,20,21]. The comparative requirements of the Ph. Eur. and ISO 3515:2002 are provided in Table 4.

The Ph. Eur. 10th edition established the limits for the composition of the essential oil obtained from the flowering tops of *L. angustifolia* for pharmaceutical use (Table 4) [13]. ISO also specifies certain characteristics of the lavender essential oils of various origins in order to facilitate the evaluation of their quality [20]. ISO 3515:2002 gives more components for the evaluation of lavender essential oils in comparison with the Ph. Eur. Furthermore, ISO also gives different acceptable ranges for lavenders from different regions. These limits vary significantly depending on origin. We intended to compare our results with the broadest ISO specification, including the combined ranges for all the origins; however, this was impossible due to contradicting regulations for lavandulol and lavandulyl acetate. For some origins, ISO determines only a minimal content and does not limit the maximal value, and for some other origins, the opposite is required. Thus, it was decided to use the “other origin” specification for the evaluation of our samples. There is no such problem while analysing the content according to the Ph. Eur., namely that there are limits without origin of lavender oil. It is worth mentioning that the Ph. Eur. sets minimal limits for 3-octanone, terpinene-4-ol, lavandulyl acetate, and lavandulol as characteristic components. According to ISO 3515:2002 (other origin), there are no lower limits required for these compounds. Moreover, for some components the limits between these two documents are relatively different, as presented in Table 4.

Table 5 and Table 6 present a comparison of the studied lavender oil composition with the acceptable ranges for the regulated components according to the Ph. Eur. and ISO norms. It is necessary to note that the percentage abundances from our experiment are semi-quantitative data obtained by GC-MS, and we cannot use them in order to conclusively evaluate the compliance with the Ph. Eur. or ISO requirements. The mentioned standards rely on values obtained by GC with a flame ionization detector (FID); however, in scientific studies, currently GC-MS is widely used for profiling and receiving quantitative sample information, and this method revolutionized the detection of minor chemical constituents in essential oils. Nevertheless, it is important to mention that FID detectors provide more accurate quantitative results for multicomponent samples because MS response factors often vary significantly [19]. Basing on the obtained GC-MS data only, we can conclude that none of the samples meet the requirements of the Ph. Eur. (Table 5) and ISO (Table 6), looking at the “other origin” specification for the chromatographic profile. The main reason for non-compliance with the pharmacopoeia monograph were isomeric terpineols: *α*-terpineol and/or terpinene-4-ol. Nonetheless, it is worth mentioning that α-terpineol and terpinen-4-ol are not considered as characteristic components of the *L. angustifolia* essential oil [12]. In addition, there are a lot of publications showing the study of the pharmacological properties of lavender oil despite the nonconformity of its chemical composition with the standards [8,18,25]. Eight samples of the essential oils out of thirteen did not meet the requirement for the terpinen-4-ol content, which ranged from 10.6% to 18.7% (cultivars 1, 3, 7, 8, 9, 12 and 13). The nine tested cultivars did not meet the requirement for *α*-terpineol (cultivars 2, 4–8, 10–12). Most samples fit the requirements of the Ph. Eur. for the linalool content, except for cultivars 4 (2018), 9, and 13. Only ten cultivars obeyed the limits for the linalyl acetate content as this content was only 7.4–19.7% in cultivars 1, 9, and 13. All of the samples with the exception of cultivar 2 met the requirement for the lavandulyl acetate (0.2–12.3%) and lavandulol (0.2–7.8%) contents. Moreover, 3-octanone was in the range of 0 to 0.95%, and cultivars 1, 2, and 4 (2018) of the essential oil did not comply with the requirement for the content of 3-octanone. Having the above-described knowledge in mind about quantitation methods (MS vs. FID), we can suppose that cultivars 3, 5, 6, 7, 8, 10, and 11 might yield an essential oil that fits the Ph. Eur. norm. It is less probable for cultivars 1, 4, 9, 12, and 13.

With regard to the ISO regulation, the first observation was that almost all of the samples did not comply with the regulated *β*-ocimenes content. The *β*-ocimenes were below the lower limit or not detected, with exception of cultivar 2 (2018). Eight samples out of thirteen are also characterized by a terpinene-4-ol content that is too high. Many samples (14 out of 19) conformed to the requirements of the ISO standard for the linalool content. In all of the studied samples, the levels of 3-octanone and camphor were in the regulated ranges. 

Regardless of the richness of the natural compounds present in essential oils, ISO and Ph. Eur. regulate only a dozen components out of hundreds. Such a targeted analysis of only a few phytochemicals is a very simplified approach, and poses a high risk of misclassifying oil quality [12]. Other non-regulated compounds contribute diversity, special character, and richness of scent notes for lavender oil, and can be ignored when looking only at ISO/Ph. Eur. GC profile compliance tests [12]. As noted by Bejar, some of the true lavender oils contain significantly different chemical compositions, and do not conform to any of the regulated specifications, even despite being authentic lavender oils [5]. The publications describe many *L. angustifolia* oils from different cultivars and origins, and a lot of them do not conform to the norms. For example, Détár et al. evaluated the chemical composition of six cultivars of *L. angustifolia* of Hungarian origin. They revealed that the linalool content was in the range of 25.7% to 55.4%. The two cultivars did not meet the requirements of the Ph. Eur. for the linalool content, as it was 50.1% and 55.4%. The linalyl acetate content was in the range of 17.7% to 42.1%. The two cultivars contained 17.7% and 18.7% of linalyl acetate, and two more cultivars contained 25.2% and 25.3% (about the lower limit). It was revealed that the *α*-terpineol content was in the range of 2.9% to 6%, meaning that none of the cultivars met the requirements of the Ph. Eur. for α-terpineol [1]. 

Lane et al. established that six samples out of ten did not meet the requirements of the Ph. Eur. for *α*-terpineol (maximum limit 2%). The content of *α*-terpineol was in the range of 0 to 5.6%. All of the samples did not meet the requirements of the Ph. Eur. for linalyl acetate (25–47%) as the main component of the *L. angustifolia* essential oil. This content ranged from 0% to 14.5%. The content of the second main component, linalool, was in the range of 17.9% to 47.8%. Three samples did not meet the requirements of the Ph. Eur. for linalool (20–45%) [22].

Our results partly conform with those of Smigielski et al., who found seventy-eight components in the Polish essential oil obtained by the hydrodistillation from dried flowers of *L. angustifolia*, cultivated in Poland. The essential oil was evaluated by GC, GC-MS, and NIR (Near InfraRed). The main components of this essential oil was linalool (30.6%), linalyl acetate (14.2%), geraniol (5.3%), *β*-caryophyllene (4.7%), and lavandulyl acetate (4.4%). Among the minor components were limonene (0.5%), camphor (0.5%), 1,8-cineole (2%), lavandulol (1.6%), and *α*-terpineol (2.7%) [7]. This essential oil did not meet the requirements of the Ph. Eur. for the linalyl acetate and α-terpineol contents.

Our results are partly in line with the results of Chen et al., where fifty compounds were identified. The main components were linalyl acetate (28.9%), linalool (24.3%), caryophyllene (7.9%), *trans*-*β*-ocimene (4.6%), 4-terpineol (4.0%), lavandulyl acetate (3.5%), borneol (2.6%), and eucalyptol (2.1%) [17]. 

Białoń et al. determined the chemical composition of the commercial *L. angustifolia* essential oil produced by ETJA (Elbląg, Poland) and the essential oil of *L. angustifolia* grown in the gardens of the Institute of Essential Oil of the Ukrainian Academy of Agricultural Sciences in Simferopol (Crimea, Ukraine). They revealed that the commercial essential oil did not meet the requirements of the Ph. Eur. for limonene (19.0% at the acceptable level of less than 1%) and lavandulyl acetate (0.06% at the acceptable level of not less than 0.2%), while the oil from the Crimean cultivar did not conform to the requirements of the Ph. Eur. for eucalyptol (5.0% at the acceptable level of less than 2.5%) and linalyl acetate (23.3% at the acceptable level of 25–47%) [11]. It is worth mentioning that the Rosea cultivar (cultivar 13) of *L. angustifolia,* grown in the gardens of the Sector of Mobilization and Conservation of Plant Resources of the Rice Institute of the National Academy of Agrarian Sciences (Ukraine) also cannot meet the requirements of the Ph. Eur. for eucalyptol (3.2% of a sum of eucalyptol and limonene at the acceptable level of less than 2.5% for eucalyptol, supposing that only eucalyptol was eluted) and linalyl acetate (7.4% at the acceptable level of 25% to 47%) as well. 

The chemical composition of essential oils from fresh and dried aerial parts and flowers of *L. angustifolia* from Wielkopolska were evaluated by Smigielski et al. Their main volatile components of the essential oils from fresh and dried flowers were linalool (26.5–34.7%), linalyl acetate (19.7–23.4%), *β*-ocimene (2.9–10.7%), *α*-terpineol (2.8–5.1%), and *α*-limonene (0.6–3.8%) [8]. Moreover, these authors studied the influence of the plant parts and the post-harvest processing procedure on the chemical composition of the essential oil. The essential oils from fresh and dried flowers contained 0.8% and 1.4% of 3-octanone; 1.5% and 0.5% of 1,8-cineole; 0.6% and 1.0% of α-limonene; 0% and 0.3% of camphor; 0.9% and 0.8% of lavandulol; 4.9% and 2.0% of terpinen-4-ol; 3.6% and 5.1% of *α*-terpineol; 34.5% and 34.7% of linalool; and 23.4% and 19.7% of linalyl acetate, respectively. On the other hand, the essential oils from fresh and dried aerial parts contained 0.9% and 0% of 3-octanone; 0.2% and 3.4% of 1,8-cineole; 3.8% and 1.2% of α-limonene; 0.1% and 0% of camphor; 0% and 0.7% of lavandulol; 4.8% and 4.5% of terpinen-4-ol; 2.9% and 2.8% of *α*-terpineol; 31.2% and 26.5% of linalool; and 23.0% and 22.5% of linalyl acetate, respectively [8]. Therefore, the evaluated cultivars of Polish origin did not meet the requirements of the Ph. Eur. and ISO for the limonene, linalyl acetate, and *α*-terpineol contents.

Dong et al. analysed the essential oil from the dried aerial parts of *L. angustifolia* collected in June 2019 in Yili (Xinjiang, China). These authors used GC-MS equipped with three capillary columns of different polarities (HP-1, HP-5ms, and HP-INNOWax) and identified forty compounds. Linalool (19.7%), linalyl acetate (26.6%), and lavandulol acetate (12.7%) were the main components of the lavender essential oil. Among the minor components were terpinene-4-ol (0.4%), camphor (0.4%), lavandulol (0.5%), and *α*-terpineol (3.6%) [4]. Thus, this essential oil also did not meet the requirements of the Ph. Eur. and ISO 3515:2002 for the content of linalool and *α*-terpineol.

The chemical composition of the essential oil, isolated from lavender (*L. angustifolia*), harvested in 2014 in Damascus Governorate (Syria), was evaluated. The essential oil was isolated by steam distillation from sample plants at the full flowering stage. The essential oil contained 35.1% of linalool, 17.7% of borneol, 14.3% of camphor, 7.6% of 1,8-cineol, 5.6% of terpinen-4-ol, and 1.5% of limonene [9]. Therefore, this essential oil did not meet the requirements of the Ph. Eur. and ISO for many indexes.

The discussed publications and our results clearly show that the chemical composition of true lavender oil is highly variable and dependent on the cultivar, origin, and many other factors. This issue creates obstacles in studying the biological effect of those oils. It is difficult to repeatedly study biological activity and reliably compare the results, as different authors use oils with very different compositions.

Some authors indicate that there is an influence of the plant growth year on the percentage content of constituents of lavender oils [1,26]. Deter et al. investigated the effect of the growth year on the *L. angustifolia* essential oil content. They concluded that the year effect was significant only in two cultivars out of the six studied. They highlighted that weather conditions had an impact on the accumulation of certain components of lavender oils [1]. We compared the influence of 2018 vs. 2017 growth years on the essential oil chemical composition of five different cultivars (4–7, 13). Some variabilities in the chemical composition were observed between the cultivars studied, and for the same cultivar cropped in different years. Figure 4 presents the change in the percentage abundance between 2018 and 2017 for the different cultivars and main oil components. Our experimental data shows no clear trend for inter-year composition changes, both regarding the studied components and the cultivars (Figure 4). A two-sided matched-pairs t-test (α = 0.05, critical t values less than −2.78 or more than 2.78) showed that we cannot reject a null hypothesis that the difference of means of the percentage abundance for years 2017 and 2018 equals zero (Table 7). Based on the statistics, it does not seem that any of the top components are characterized by a significant increase or decrease in percentage abundance across different cultivars. 

The biggest variability was seen in cultivars 4 and 7 (Table 3, Figure 4). It relates both to the content of the main two components, as well as for minor ones. For example, the main oil component in 2017 was linalool for cultivar 7, but linalyl acetate was the main oil component in 2018. There are two possible reasons for such changes: the influence of the year and the vegetation phase (full flowering and the end of full flowering). It seems that this cultivar does not keep the stability for the regulated components. Regarding cultivar 4, linalyl acetate was the most prevalent compound (44.2%), which was followed by linalool 25.7% and *α*-terpineol (4.5%) in 2017. Their percentage content decreased in 2018 compared to 2017. It seems that this cultivar also did not show stability concerning the principal major and minor essential oil components. Some inter-year variations were also present for cultivar 13, especially for linalool and terpinene-4-ol. It seems that this cultivar maintains composition stability during these two years for some other key components: linalyl acetate, camphor, and lavandulyl acetate. Moreover, as discussed earlier, this cultivar is characterized by a significantly low level of linalyl acetate and a too-high linalool level considering the norms, and even compared to other cultivars tested. Furthermore, it seems that the composition of the essential oils from cultivars 5 and 6 maintains inter-year stability. Big changes in the chemical composition of the lavender oil grown on the same plot and for the same cultivar is not a good feature, as it is not welcome by lavender growers due to unpredictable crop oil properties.

It needs to be highlighted that the oils of four cultivars (5, 6, 7, and 13 in 2018) were distilled from flowers harvested at the end of full flowering. According to some publications, the phenological stage influences the composition of the essential oil [27,28,29]. Hassiotis et al. studied different factors on lavender oil production and its quality, and showed that the later flowering stages were beneficial for linalool production. Thus, also in our study, it might add some additional factor that created variation between the studied years. Unfortunately, due to insufficient sample availability, this effect could not be analysed in this study. The comparative analysis of the samples from the end of flowering versus the full flowering vegetation phase was not possible, because inter-cultivar variations are big, and probably much bigger than the above variations. Table 3 shows that the inter-year changes for the content of the main oil components are not bigger for the cultivars with different flowering stages than for the cultivars with the same flowering stage. Our late-flowering samples (except cultivar 13) do not characterize a higher linalool content, as observed by Hassiotis. Nevertheless, our experiments do not provide substantial data to analyse the factor of the flowering phase, and more experiments are needed to elucidate its effect.

Taking into account the morphological features, oil yields (presented in Table 8), the oil composition (the results presented in this paper), and the other data gathered by the Rice Institute (Figure 5), we consider the following suitability of tested cultivars:Cultivar 1 (2-15) can be used in landscaping. According to the Ph. Eur., this cultivar is not suitable for pharmaceutical purposes because of its nonconformity for the content of the major characteristic components;Cultivar 2 (Victoria) could be utilized for the production of essential oil, including for pharmaceutical purposes without taking the α-terpineol content into consideration;Cultivar 3 (1-4-09) can be used for selective breeding;Cultivar 4 (Lidia) could be used in the production of essential oil for use in perfumery;Cultivars 5 (1-3-16), 6 (1-2-16), and 11 (701-2) can be used in selective breeding and for the production of essential oil;Cultivars 7 (Alba), 9 (2-2-3), 12 (21-19), and 13 (Rosea) can be used in selective breeding and landscaping;Cultivars 8 (2-1-17) and 10 (2-4-6) can be used in the production of essential oil and landscaping.

It is worth noting that some of the above-described cultivars are already successfully cropped in Ukraine (cultivars 5–7, 9, 11, 12, 13).

Our study has some drawbacks. The first drawback is that the sample collection was not always done from the same exact vegetation phase: full flowering and the end of full flowering (Table 8). The other drawback is related to the use of only GC-MS, and no GC-FID data availability. On the other hand, this study has a lot of advantages. First of all, the essential oils of numerous new lavender cultivars, which had been grown on the same plot, were evaluated. Secondly, some cultivars were evaluated from the point of view of the influence of the year. In addition, all of the samples were analysed by the same operator using the same method and instrument.

## 3. Materials and Methods

### 3.1. Plant Material

Flowering parts of *L. angustifolia* were collected in the flowering stage in Kherson region (Ukraine) in 2016, 2017, and 2018. The voucher specimens of each year were deposited at the Herbarium of the Sector of Mobilization and Conservation of Plant Resources of the Rice Institute of the NAAS (Plodove, Kherson region). All of the oil yields were calculated for a fresh mass. One-hundred grams of flowering parts of *L. angustifolia* were used for each distillation. The detailed information about the tested cultivars is provided in Table 8.

### 3.2. Essential Oil Isolation Procedure

The essential oils from fresh flowers of *L. angustifolia* were extracted via hydrodistillation for 4 h with a Clevenger-type apparatus. The oils were kept at room temperature in sealed tubes (protected from light) before analysis by GC-MS.

### 3.3. GC-MS Analysis of the Essential Oil

The volatile compounds of the *L. angustifolia* were identified by comparing the mass spectra data with the spectrometer database of the NIST 11 Library, and by comparison of their retention index calculated against *n*-alkanes (C_9_–C_20_). Each chromatographic analysis was repeated three times. The average value of the relative composition of the essential oil percentage was calculated from the peak areas. The Hewlett Packard HP 6890 series GC system chromatograph was used for the study, which was coupled with the Hewlett Packard 5973 mass selective detector. The GC column used was a non-polar, high-temperature ZB-5HT (5% diphenyl- and 95% dimethylpolysiloxane) with a capillary column that was 30 m long, an inner diameter of 0.32 mm, and a film thickness of 0.25 μm (Phenomenex Inc., Torrance, CA, USA). The gas chromatograph was equipped with a split injector; the split ratio was 20:1 and 1 μm of a sample was introduced. The injector temperature was 250 °C. Helium served as the carrier gas, and its flow rate was 2 mL/min. The oven program was 40–180 °C with the heating rate 5 °C/min, and 180–280 °C with the heating rate of 10 °C/min. The essential oil sample (20 µL) was dissolved in 1 mL of dichloromethane and directly analysed. The relative amounts of the identified (and few unidentified) components represent the percentage abundance (area percent, solvent peak excluded). 

### 3.4. Statistical Analysis

All of the analyses were carried out in triplicate, and the results were expressed as mean. To present large datasets, standard deviations were presented only in a large Appendix A. To investigate inter-year variability and find any trends for changing content of the main oil components, a two-sided matched pairs *t*-test was conducted with α = 0.05, using the Excel data analysis ToolPak (Office 2019).

## 4. Conclusions

Firstly, the chemical composition of lavender oils from 13 new cultivars of Ukrainian origin were analysed. It was revealed that linalool and linalyl acetate were the principal components of most of the lavender oils. The oils did not conform to the requirements of the Ph. Eur. and ISO 5315:2002 for the chemical composition of *L. angustifolia*. The main reason for such nonconformity is α-terpineol and/or terpinene-4-ol. Secondly, despite the same plot of growing, the chemical composition of some cultivars differed from year to year, which might be due to the specific weather conditions. Thirdly, the obtained results will help to more widely characterise the new cultivars regarding their use in breeding, landscaping, and essential oil production for different industries, including the pharmaceutical, cosmetic, and perfume industries. 

Taking into account our results and reviewed literature data, it seems that it is worth considering changes of the specification norms for the chromatographic profile, especially for terpineol isomers. Numerous publications point out that the lavender oil quality has no strict connection with phytochemicals regulated by ISO and Ph. Eur., and such specification ranges to a certain degree are arbitrary, especially for ISO, where the different origins have distinctive limits. 

## Figures and Tables

**Figure 1 molecules-26-05681-f001:**
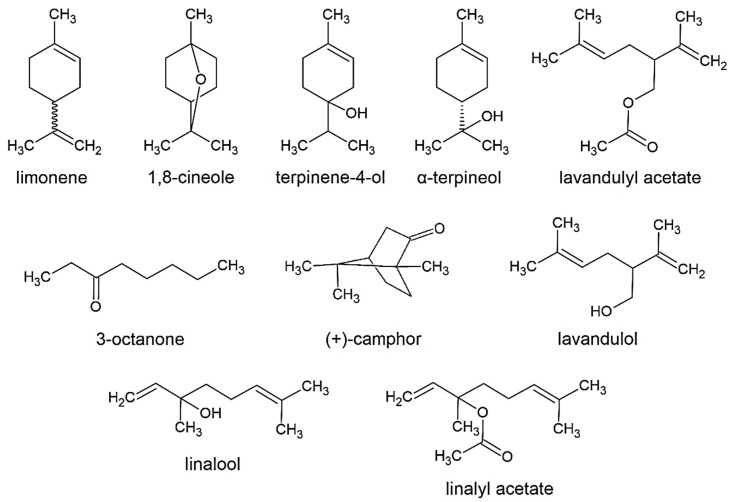
Structures of the characteristic components of essential oils of *L. angustifolia*.

**Figure 2 molecules-26-05681-f002:**
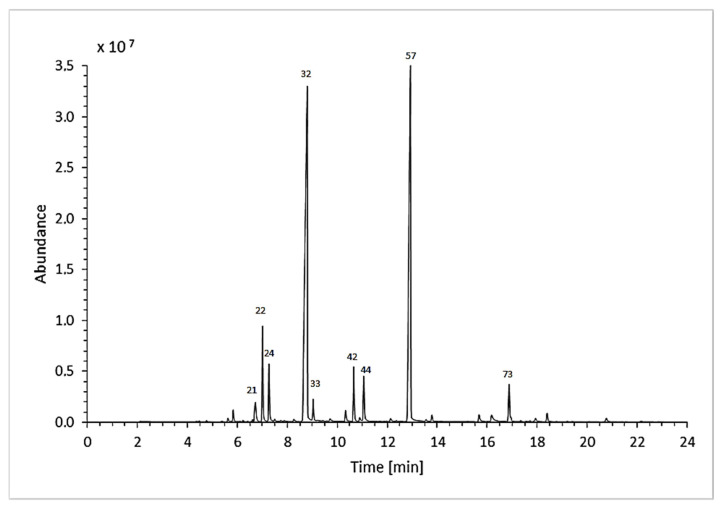
Exemplary chromatogram (GC-MS, TIC) of lavender essential oil. The sample analysed below was obtained from cultivar 2 grown in 2018. The numbers are related to the numeration used in Table 1 and Table 2.

**Figure 3 molecules-26-05681-f003:**
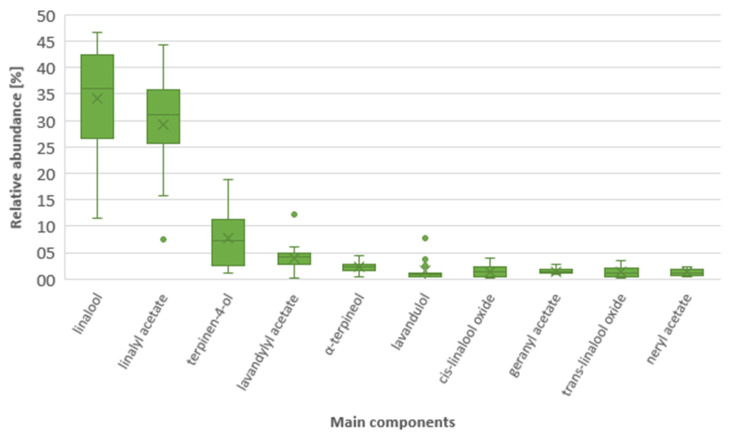
Relative abundance of the main components in the studied lavender essential oils. The box section represents the interquartile range (the results between quartile Q1 and Q3). A straight line intersecting the box represents the median, and a cross represents the average value. Whiskers represent the lowest and highest results, and separate points represent outlying values.

**Figure 4 molecules-26-05681-f004:**
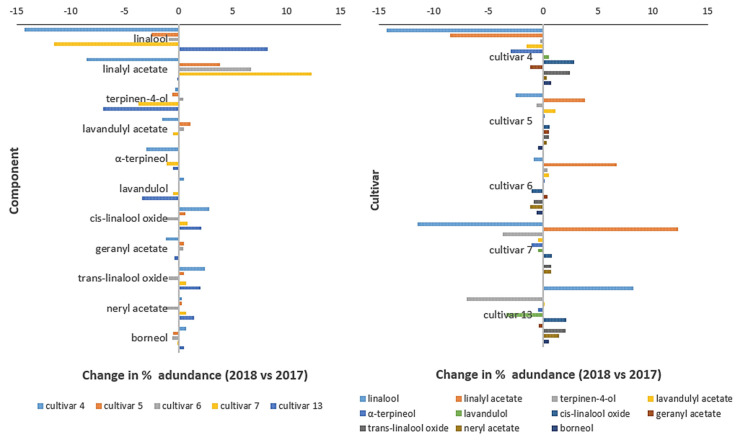
Inter-year (2018 vs. 2017) changes of relative % abundancies of the main oil components cropped from the different studied lavender cultivars. The left plot highlights changes for oil components and the right plot highlights changes for cultivars.

**Figure 5 molecules-26-05681-f005:**
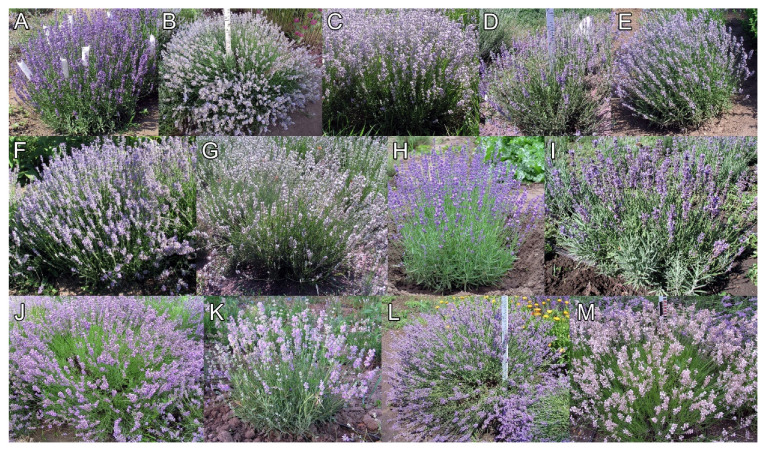
Illustrative photos for the specimens of different studied lavender cultivars: (**A**) 2-15 (1); (**B**) Victoria (2); (**C**) 1-4-09 (3); (**D**) Lidia (4); (**E**) 1-3-16 (5); (**F**) 1-4-09 (6); (**G**) Alba (7); (**H**) 2-1-17 (8); (**I**) 2-2-3 (9); (**J**) 2-4-6 (10); (**K**) 701-2 (11); (**L**) 21-19 (12); (**M**) Rosea (13).

**Table 1 molecules-26-05681-t001:** GC data for essential oil components identified in the samples from the different *L. angustifolia* cultivars (1–6). Apart from LRI (experimental and reference values), all the figures represent % abundance (area percent without solvent peak). * tentative identification.

Component		LRI		Cultivar									
				1	2		3	4		5		6	
		Exp.	Ref.	2016	2016	2018	2016	2017	2018	2017	2018	2017	2018
1	1-hexanol	869	870										
2	tricyclene	923	923					0.02		0.04		0.05	
3	*α*-thujene	928	928			0.05		0.02		0.05		0.03	
4	*α*-pinene	935	936			0.06		0.09		0.24		0.14	
5	camphene	950	950			0.08		0.38		0.52	0.03	0.61	0.19
6	sabinene	975	973			0.07		0.01		0.02		0.01	
7	*β*-pinene	977	978					0.06		0.06		0.04	
8	1-octen-3-ol	982	980	0.27	0.42	0.23	0.09	0.07		0.19		0.10	
9	3-octanone	987	985	0.09	0.06		0.26	0.25		0.32	0.06	0.81	0.22
10	*β*-myrcene	991	989	0.03	0.05	0.63		0.35	0.14	0.58	0.18	0.14	0.26
11	3-octanol	995	993	0.17	0.02		0.23	0.06		0.19		0.21	
12	butyl butanoate	996	997								0.08		
13	*α*-phellandrene	1004	1004			0.03							
14	2-carene	1005	1003			0.10							
15	3-carene	1010	1011		0.01	0.04		0.03		0.15			
16	hexyl acetate	1014	1010	0.07			0.22	0.24		0.44	0.20	0.44	0.21
17	*p*-cymene	1026	1026			0.11		0.05	0.21	0.06	0.13	0.04	0.34
18	*o*-cymene	1041	1041	0.09	0.19	0.03	0.19	0.28		0.12		0.23	
19	limonene	1031	1030			0.58						0.16	0.14
20	limonene-eucalyptol coelution	1032	1032	0.22	0.86		0.20	0.90	0.11	1.18	0.40		

21	eucalyptol	1032	1032			1.14						0.09	0.10
22	*cis*-*β*-ocimene	1040	1038		0.01	4.73		0.01	0.34	0.44	0.02		0.03
23	lavender lactone	1044	1039	0.05			0.05					0.07	
24	*trans*-*β*-ocimene	1051	1048		0.03	2.96		0.02		1.00	0.04		0.04
25	*γ*-terpinene	1061	1060		0.01	0.12		0.02		0.09			
26	sabinene hydrate*	1062	1065	0.58	0.01	0.08	0.67	0.04		0.10	0.06	0.07	0.07
27	*cis*-linalool oxide	1075	1075	2.70	1.30	0.07	2.31	1.11	3.85	0.37	0.97	2.79	1.74
28	1-octanol	1078	1072										
29	camphenilone	1083	1085							0.02			
30	*trans*-linalool oxide	1089	1083	2.55	1.22	0.19	2.12	0.97	3.42	0.39	0.89	2.50	1.61
31	*α*-terpinolene *	1094	1091										
32	linalool	1102	1099	43.87	44.05	40.75	29.20	25.68	11.42	42.37	39.85	36.10	35.20
33	1-octen-3-yl-acetate	1114	1110	0.48	1.95	1.14	0.67	1.30	0.51	1.31	0.87	1.02	0.54
34	*cis*-pinene hydrate	1122	1121										
35	*cis*-2-menthenol	1127	1121										
36	3-octanol, acetate	1134	1131				0.22					0.12	
37	*trans*-pinocarveol	1142	1140										
38	camphor	1148	1143	0.42	0.43	0.20	0.48	0.47	0.43	0.63	0.73	0.77	0.60
39	heptyl propionate	1164	1169					0.06				0.05	
40	borneol	1169	1166	2.07	0.99	0.72	1.79	1.58	2.30	1.95	1.47	2.11	1.45
41	lavandulol	1172	1168	2.61	0.26	0.03	0.68	0.35	0.91	0.97	1.04	0.78	0.84
42	terpinen-4-ol	1179	1177	10.55	3.81	3.07	11.25	1.51	1.17	3.10	2.49	2.17	2.64
43	cryptone	1185	1184	0.30	0.32	0.28							
44	*α*-terpineol	1193	1190	1.00	4.12	2.82	1.95	4.52	1.48	2.27	2.39	2.65	2.82
45	2,6-dimethyl-3,7-octadiene-2,6-diol	1195	1190				0.77		2.33		1.07		1.38

46	hexyl butanoate	1208	1191	1.27			0.38						
47	carveol	1212	1219	0.16	0.45		0.20	0.44		0.45		1.08	
48	bornyl formate/isobornyl formate *	1229	1229	1.15				0.10	0.51	0.07	0.16	0.27	0.30
49	*cis*-geraniol (nerol)	1232	1229		0.40	0.24	0.12	0.43		0.27	0.26	0.25	0.25
50	cumin aldehyde	1239	1238										
51	isothymol methyl ether	1238	1244										
52	carvone	1244	1242										
53	thymol methyl ether	1246	1234										
54	*β*-citral (=neral)	1247	1242				1.64	0.15			0.10	0.10	
55	unidentified	1249											
56	unidentified	1255							0.23		0.17	0.13	0.26
57	linalyl acetate	1259	1255	15.79	31.11	33.81	26.38	44.20	35.72	31.05	34.88	28.61	35.27
58	*α*-citral (=geranial)	1278	1270								0.15		0.25
59	unidentified	1274	1272	0.39	0.27		0.60	0.38				0.28	
60	unidentified	1276		0.31			0.31					0.17	
61	2,6-dimethyl-1,7-octadiene-3,6-diol	1284	1286										

62	bornyl acetate	1287	1284		0.16	0.11	0.34	0.96	1.29	0.55	0.72	0.85	0.97
63	lavandulyl acetate	1291	1289	4.02	0.20	0.39	3.68	4.22	2.68	4.88	5.97	4.72	5.15
64	thymol	1296	1290										
65	lavandulyl propionate	1302			0.08			0.12					
66	linalyl propionate	1332	1336					0.06					
67	unidentified	1338		0.18	0.14		0.65	0.20	4.93	0.04	0.30	0.50	0.65
68	unidentified	1344		0.49	0.10		1.18	0.67	6.25		0.38	1.01	0.91
69	unidentified	1354		0.51	0.10		1.14	0.73	6.16		0.49	1.02	1.06
70	neryl acetate	1365	1363	1.99	1.26	0.56	2.23	1.79	2.10	0.54	0.80	2.33	1.07
71	geranyl acetate	1384	1380	0.85	1.87	0.77	1.64	2.65	1.54	0.95	1.48	1.42	1.81
72	*β*-elemene	1393	1390				0.54						
73	caryophyllene	1423	1420	0.05	0.56	2.46	0.13	0.06		0.34		0.06	
74	*α*-santalene	1424	1421	0.31	0.23		0.21	0.07		0.06		0.09	
75	*trans*-*α*-bergamotene	1438	1434	0.06	0.07	0.07							
76	humulene	1457	1453			0.07							
77	*trans*-*β*-farnesene	1459	1456	0.41	0.12	0.31	0.56	0.13		0.48	0.26	0.09	
78	alloaromadendrene	1465	1460										
79	germacrene D	1484	1481			0.55	0.35			0.12		0.07	
80	*α*-farnesene	1489	1491										
81	unidentified	1499		0.52	0.17		0.44	0.44				1.41	
82	*β*-bisabolene*	1510	1508						8.23				0.46
83	neryl propionate *	1510						0.95		0.19		0.11	
84	*γ*-cadinene	1516	1513	0.21	0.07		1.64				0.14		
85	*δ*-cadinene	1522	1523	0.10	0.11		0.27						
86	*cis*-2-pentadecen-4-yne *	1536		0.09			0.13					0.20	
87	caryophyllene oxide	1589	1581	1.90	2.33	0.30	1.82	0.16	1.23	0.06	0.39	0.47	0.84
88	ledol (=globulol)	1590	1582										
89	humulene oxide II	1615	1606		0.05		0.04	0.19		0.06		0.04	
90	*α*-muurolol	1642	1640			0.08			0.53		0.38		0.32
91	unidentified	1647		0.77	0.07		0.05	0.20		0.72		0.28	

**Table 2 molecules-26-05681-t002:** GC data for the essential oil components identified in the samples from the different *L. angustifolia* cultivars (7–13). Apart from LRI (experimental and reference values), all the figures represent % abundance (area percent without solvent peak). * tentative identification.

Component		LRI		Cultivar								
				7		8	9	10	11	12	13	
		Exp.	Ref.	2017	2018	2017	2017	2017	2017	2017	2017	2018
1	1-hexanol	869	870								0.30	0.16
2	tricyclene	923	923									
3	*α*-thujene	928	928	0.02		0.04	0.36	0.04	0.04	0.21	0.15	0.02
4	*α*-pinene	935	936	0.05		0.06	0.51	0.12	0.06	0.39	0.26	0.05
5	camphene	950	950	0.01		0.02	0.18	0.06	0.04	0.05	0.03	
6	sabinene	975	973	0.04		0.02	0.13	0.04	0.01	0.07	0.06	
7	*β*-pinene	977	978	0.19	0.06	0.02	0.08	0.04	0.01	0.15	0.09	
8	1-octen-3-ol	982	980	0.15	0.03	0.09	0.28	0.07	0.88	0.08	0.85	0.74
9	3-octanone	987	985	0.95	0.35	0.45	0.91	0.65	0.10	0.84	0.72	0.27
10	*β*-myrcene	991	989	0.36	0.15	0.40	0.31	0.13	0.18	0.51	0.33	0.06
11	3-octanol	995	993	0.60		0.17	0.49	0.31	0.04	0.55	0.25	
12	butyl butanoate	996	997		0.25							0.10
13	*α*-phellandrene	1004	1004									
14	2-carene	1005	1003									
15	3-carene	1010	1011	0.04		0.08	0.30		0.07	0.12	0.15	
16	hexyl acetate	1014	1010	0.95	0.45	0.20	1.19	0.61	0.02	0.50	1.20	0.44
17	*p*-cymene	1026	1026	0.03	0.40	0.04	0.10		0.05	0.12		0.58
18	*o*-cymene	1041	1041	0.55	0.05	0.39	1.39	0.51	0.27	0.84	1.08	
19	limonene	1031	1030	0.18				0.14				0.14
20	limonene-eucalyptol coelution	1032	1032		0.15	0.68	1.31		0.80	0.89	3.20	
21	eucalyptol	1032	1032	0.11				0.11				0.31
22	*cis*-*β*-ocimene	1040	1038	0.02	0.02	0.03			0.02	0.02	0.13	0.06
23	lavender lactone	1044	1039	0.02	0.05			0.06			0.02	
24	*trans*-*β*-ocimene	1051	1048	0.04	0.03	0.04			0.07	0.10	0.05	
25	*γ*-terpinene	1061	1060	0.03							0.03	
26	sabinene hydrate *	1062	1065	0.18	0.26	0.26	0.79	0.31	0.12	0.35	0.58	0.51
27	*cis*-linalool oxide	1075	1075	0.81	1.61	0.45	0.73	1.81	1.51	0.53	0.53	2.56
28	1-octanol	1078	1072				0.06		0.06		0.07	
29	camphenilone	1083	1085									
30	*trans*-linalool oxide	1089	1083	0.69	1.43	0.42	0.62	1.58	1.35	0.49	0.48	2.48
31	*α*-terpinolene*	1094	1091			0.14	0.44					
32	linalool	1102	1099	38.19	26.66	28.88	19.81	23.27	44.13	34.06	38.54	46.74
33	1-octen-3-yl-acetate	1114	1110	0.92	0.77	1.42	1.37	0.77	1.85	0.47	0.24	0.21
34	*cis*-pinene hydrate	1122	1121		0.19							
35	*cis*-2-menthenol	1127	1121				0.11			0.03	0.18	
36	3-octanol, acetate	1134	1131	0.31		0.25	0.08	0.16		0.27		
37	trans-pinocarveol	1142	1140	0.28	0.18			0.13				
38	camphor	1148	1143	0.36	0.22	0.10	0.38	0.22	0.38	0.69	0.21	0.17
39	heptyl propionate	1164	1169	0.09			0.17	0.12		0.07	0.17	
40	borneol	1169	1166	1.30	1.22	0.48	1.44	0.66	1.17	1.17		0.47
41	lavandulol	1172	1168	0.85	0.44	0.73	3.57	0.86	0.22	0.88	7.81	4.39
42	terpinen-4-ol	1179	1177	10.99	7.28	10.73	18.73	7.63	2.86	18.33	18.61	11.60
43	cryptone	1185	1184				0.69		0.08	0.10	0.81	0.39
44	*α*-terpineol	1193	1190	3.54	2.40	3.26	1.25	2.48	2.44	2.54	0.97	0.50
45	2,6-dimethyl-3,7-octadiene-2,6-diol	1195	1190		1.14						1.29	2.47
46	hexyl butanoate	1208	1191								0.59	
47	carveol	1212	1219	1.20		0.35	1.71	0.94	0.28	0.79		
48	bornyl formate/isobornyl formate*	1229	1229	0.04	0.16		0.27	0.14	0.05	0.06		0.24
49	*cis*-geraniol (nerol)	1232	1229	0.44	0.20	0.33		0.13	0.23	0.25		0.14
50	cumin aldehyde	1239	1238									0.18
51	isothymol methyl ether	1238	1244	0.14								
52	carvone	1244	1242				0.35			0.04	0.58	
53	thymol methyl ether	1246	1234	0.10								
54	*β*-citral (=neral)	1247	1242	0.17	0.11				0.04	0.05		
55	unidentified	1249		0.41		0.39	0.89	1.19	0.10	0.57		
56	unidentified	1255			1.02							1.17
57	linalyl acetate	1259	1255	25.68	37.99	39.26	19.71	38.00	34.26	26.23	7.48	7.39
58	*α*-citral (=geranial)	1278	1270		0.56							0.69
59	unidentified	1274	1272	0.29	0.22			0.42	0.09	0.10		
60	unidentified	1276										
61	2,6-dimethyl-1,7-octadiene-3,6-diol	1284	1286									0.53
62	bornyl acetate	1287	1284		0.25			0.49	0.05			0.25
63	lavandulyl acetate	1291	1289	2.91	2.43	4.72	12.25	4.26	0.32	3.00	4.83	4.94
64	thymol	1296	1290	0.82								
65	lavandulyl propionate	1302										
66	linalyl propionate	1332	1336					0.21				
67	unidentified	1338		0.11	0.80			0.62	0.15	0.03		0.46
68	unidentified	1344		0.15	1.39	0.20	0.23	0.08		0.09		0.28
69	unidentified	1354		0.16	1.36	0.23	0.28	0.10		0.11		0.28
70	neryl acetate	1365	1363	1.12	1.80	0.83	0.57	1.93	0.83	0.68	0.50	1.86
71	geranyl acetate	1384	1380	1.85	1.85	1.27	1.32	1.38	0.83	1.01	1.22	0.81
72	*β*-elemene	1393	1390									0.77
73	caryophyllene	1423	1420	0.30		0.77	0.41	0.07	0.63	0.42	1.29	0.08
74	*α*-santalene	1424	1421	0.04		0.13	0.20	0.11	0.23	0.06		0.25
75	*trans*-*α*-bergamotene	1438	1434						0.07			
76	humulene	1457	1453									
77	*trans*-*β*-farnesene	1459	1456	0.32	0.31	0.29	0.80	0.39	0.12	0.37		0.40
78	alloaromadendrene	1465	1460									
79	germacrene D	1484	1481									
80	*α*-farnesene	1489	1491								1.70	0.66
81	unidentified	1499							0.26	0.13		
82	*β*-bisabolene*	1510	1508		2.00							0.28
83	neryl propionate*	1510					0.21				0.17	
84	*γ*-cadinene	1516	1513									
85	*δ*-cadinene	1522	1523									
86	*cis*-2-pentadecen-4-yne *	1536						0.36	0.07	0.04		
87	caryophyllene oxide	1589	1581	0.81	1.64	1.03	2.26	0.75	1.65	0.57	1.63	2.20
88	ledol (=globulol)	1590	1582									0.49
89	humulene oxide II	1615	1606						0.04			
90	*α*-muurolol	1642	1640		0.13							0.24
91	unidentified	1647		0.10		0.36	0.75	0.11	0.04		0.60	

**Table 3 molecules-26-05681-t003:** Top ten components in the essential oils of the studied lavenders cultivars. All of the figures represent % abundance. The components are listed in descending order of the sum of % abundance for each component.

Rank	Compound	Cultivar and Year of Cultivation																		
		1	2		3	4		5		6		7		8	9	10	11	12	13	
		2016	2016	2018	2016	2017	2018	2017	2018	2017	2018	2017	2018	2017	2017	2017	2017	2017	2017	2018
1	linalool	43.9	44.1	40.8	29.2	25.7	11.4	42.4	39.9	36.1	35.2	38.2	26.7	28.9	19.8	23.3	44.1	34.1	38.5	46.7
2	linalyl acetate	15.8	31.1	33.8	26.4	44.2	35.7	31.1	34.9	28.6	35.3	25.7	38.0	39.3	19.7	38.0	34.3	26.2	7.5	7.4
3	terpinen-4-ol	10.6	3.8	3.1	11.3	1.5	1.2	3.1	2.5	2.2	2.6	11.0	7.3	10.7	18.7	7.6	2.9	18.3	18.6	11.6
4	lavandulyl acetate	4.0	0.2	0.4	3.7	4.2	2.7	4.9	6.0	4.7	5.2	2.9	2.4	4.7	12.3	4.3	0.3	3.0	4.8	4.9
5	*α*-terpineol	1.0	4.1	2.8	2.0	4.5	1.5	2.3	2.4	2.7	2.8	3.5	2.4	3.3	1.3	2.5	2.4	2.5	1.0	0.5
6	lavandulol	2.6	0.3	0.0	0.7	0.4	0.9	1.0	1.0	0.8	0.8	0.9	0.4	0.7	3.6	0.9	0.2	0.9	7.8	4.4
7	*cis*-linalool oxide	2.7	1.3	0.1	2.3	1.1	3.9	0.4	1.0	2.8	1.7	0.8	1.6	0.5	0.7	1.8	1.5	0.5	0.5	2.6
8	geranyl acetate	0.9	1.9	0.8	1.6	2.7	1.5	1.0	1.5	1.4	1.8	1.9	1.9	1.3	1.3	1.4	0.8	1.0	1.2	0.8
9	*trans*-linalool oxide	2.6	1.2	0.2	2.1	1.0	3.4	0.4	0.9	2.5	1.6	0.7	1.4	0.4	0.6	1.6	1.4	0.5	0.5	2.5
10	neryl acetate	2.0	1.3	0.6	2.2	1.8	2.1	0.5	0.8	2.3	1.1	1.1	1.8	0.8	0.6	1.9	0.8	0.7	0.5	1.9
% of total oil composition		86	89	83	82	87	64	87	91	84	88	87	84	91	79	83	89	88	81	83

**Table 4 molecules-26-05681-t004:** The comparative requirements of Ph. Eur. (10th edition) and ISO 3515:2002 for the essential oil of * L. angustifolia*.

Component	Requirements	
	Ph. Eur.	ISO 3515:2002 (Other Origin) ^a^
limonene	≤1%	≤1%
1.8-cineole ^b^	≤2.5%	≤3%
*β*-phellandrene ^b^	–	≤1%
*cis*-*β*-ocimene	–	1–10%
*trans*-*β*-ocimene	–	0.5–6%
3-octanone	0.1–5%	≤3%
camphor	≤1.2%	≤1.5%
linalool	20–45%	20–43%
linalyl acetate	25–47%	25–47%
terpinene-4-ol	0.1–8%	≤8%
lavandulyl acetate	min. 0.2%	≤8%
lavandulol	min. 0.1%	≤3%
*α*-terpineol	≤2%	≤2%

^a^ ISO provides different specifications depending on origin. ^b^ 1.8-cineole and β-phellandrene can be coeluted.

**Table 5 molecules-26-05681-t005:** Comparative table with % relative abundancies and normative content values given by Ph. Eur 9.

	Ph. Eur. Norm (%)	Cultivars and Year of Cultivation																		
Regulated Component		1	2		3	4		5		6		7		8	9	10	11	12	13	
		2016	2016	2018	2016	2017	2018	2017	2018	2017	2018	2017	2018	2017	2017	2017	2017	2017	2017	2018
limonene	≤1			0.58						0.16	0.14	0.18				0.14				0.14
eucalyptol	≤2.5			1.14						0.09	0.10	0.11				0.11				0.31
limonene+eucalyptol		0.22	0.86		0.20	0.90	0.11	1.18	0.40				0.15	0.68	1.31		0.80	0.89	3.20	
3-octanone	0.1–5	0.09	0.06	nd	0.26	0.25	nd	0.32	0.06	0.81	0.22	0.95	0.35	0.45	0.91	0.65	0.10	0.84	0.70	0.27
linalool	20–45	43.87	44.05	40.75	29.20	25.68	11.42	42.37	39.85	36.10	35.20	38.19	26.66	28.88	19.81	23.27	44.13	34.06	38.50	46.74
camphor	≤1.2	0.42	0.43	0.20	0.48	0.47	0.43	0.63	0.73	0.77	0.60	0.36	0.22	0.10	0.38	0.22	0.38	0.69	0.20	0.17
lavandulol	min. 0.1	2.61	0.26	0.03	0.68	0.35	0.91	0.97	1.04	0.78	0.84	0.85	0.44	0.73	3.57	0.86	0.22	0.88	7.80	4.39
terpinen-4-ol	0.1–8	10.55	3.81	3.07	11.25	1.51	1.17	3.10	2.49	2.17	2.64	10.99	7.28	10.73	18.73	7.63	2.86	18.33	18.60	11.60
*α*-terpineol	≤2	1.00	4.12	2.82	1.95	4.52	1.48	2.27	2.39	2.65	2.82	3.54	2.40	3.26	1.25	2.48	2.44	2.54	1.00	0.50
linalyl acetate	25–47	15.79	31.11	33.81	26.38	44.20	35.72	31.05	34.88	28.61	35.27	25.68	37.99	39.26	19.71	38.00	34.26	26.23	7.48	7.39
lavandulyl acetate	min. 0.2	4.02	0.20	0.39	3.68	4.22	2.68	4.88	5.97	4.72	5.15	2.91	2.43	4.72	12.25	4.26	0.32	3.00	7.50	4.94

Note: Due to a coelution of limonene and eucalyptol, they are presented as a sum. In some cases, it was possible to quantitate them separately. nd; not detected.

**Table 6 molecules-26-05681-t006:** Comparative table with % relative abundancies and normative content values given by ISO 3515 (other origins).

	ISO 3515 Norm (%)	Cultivars and Year of Cultivation																		
Regulated Components		1	2		3	4		5		6		7		8	9	10	11	12	13	
		2016	2016	2018	2016	2017	2018	2017	2018	2017	2018	2017	2018	2017	2017	2017	2017	2017	2017	2018
limonene	≤1			0.58						0.16	0.14	0.18				0.14				0.14
eucalyptol	≤3			1.14						0.09	0.10	0.11				0.11				0.31
limonene+eucalyptol	max. 4	0.22	0.86		0.20	0.90	0.11	1.18	0.40				0.15	0.68	1.31		0.80	0.89	3.20	
*cis*-*β*-ocimene	1–10	nd	0.01	4.73	nd	0.01	0.34	0.44	0.02	nd	0.03	0.02	0.02	0.03	nd	nd	0.02	0.02	0.13	0.06
*trans*-*β*-ocimene	0.5–6	nd	0.03	2.96	nd	0.02	nd	1.00	0.04	nd	0.04	0.04	0.03	0.04	nd	nd	0.07	0.10	0.05	nd
3-octanone	≤3	0.09	0.06	Nd	0.26	0.25	nd	0.32	0.06	0.81	0.22	0.95	0.35	0.45	0.91	0.65	0.10	0.84	0.70	0.27
camphor	≤1.5	0.42	0.43	0.20	0.48	0.47	0.43	0.63	0.73	0.77	0.60	0.36	0.22	0.10	0.38	0.22	0.38	0.69	0.20	0.17
linalool	20–43	43.87	44.05	40.75	29.20	25.68	11.42	42.37	39.85	36.10	35.20	38.19	26.66	28.88	19.81	23.27	44.13	34.06	38.50	46.74
linalyl acetate	25–47	15.79	31.11	33.81	26.38	44.20	35.72	31.05	34.88	28.61	35.27	25.68	37.99	39.26	19.71	38.00	34.26	26.23	7.48	7.39
lavandulol	≤3	2.61	0.26	0.03	0.68	0.35	0.91	0.97	1.04	0.78	0.84	0.85	0.44	0.73	3.57	0.86	0.22	0.88	7.80	4.39
terpinen-4-ol	≤8	10.55	3.81	3.07	11.25	1.51	1.17	3.10	2.49	2.17	2.64	10.99	7.28	10.73	18.73	7.63	2.86	18.33	18.60	11.60
lavandulyl acetate	≤8	4.02	0.20	0.39	3.68	4.22	2.68	4.88	5.97	4.72	5.15	2.91	2.43	4.72	12.25	4.26	0.32	3.00	7.50	4.94
*α*-terpineol	≤2	1.00	4.12	2.82	1.95	4.52	1.48	2.27	2.39	2.65	2.82	3.54	2.40	3.26	1.25	2.48	2.44	2.54	1.00	0.50

Note: Due to a coelution of limonene and eucalyptol, they are presented as a sum. *β*-phellandrene was not detected in studied samples, but its traces might coelute with limonene and eucalyptol. nd; not detected.

**Table 7 molecules-26-05681-t007:** The results of paired *t*-test (two-sided) for top lavender oil component change trends between 2018 and 2017 from five different cultivars. The sample size for statistics was 5 (n = 5), and the degrees of freedom df = 4. The null hypothesis was that the content difference between the years is equal to 0. The alternative hypothesis was that it is not equal to zero.

Compound	Average Content (%)		Variance (%^2^)		t Statistics	t Critical Value for α = 0.05 and df = 4
	2017	2018	2017	2018		
linalool	36.18	31.98	39.50	185.30	1.05	2.78
linalyl acetate	27.42	30.26	174.07	164.75	−0.82	
terpinen-4-ol	7.28	5.04	54.67	18.82	1.62	
lavandulyl acetate	4.30	4.24	0.69	2.55	0.13	
*α*-terpineol	2.80	1.92	1.72	0.86	1.53	
lavandulol	2.18	1.50	9.92	2.68	0.97	
*cis*-linalool oxide	1.12	2.16	0.96	1.27	−1.55	
geranyl acetate	1.64	1.50	0.46	0.19	0.45	
*trans*-linalool oxide	1.02	1.96	0.74	0.98	−1.60	
neryl acetate	1.24	1.54	0.64	0.31	−0.71	
borneol	1.40	1.40	0.72	0.42	0.00	

**Table 8 molecules-26-05681-t008:** Information about the tested cultivars.

No	Cultivar Name	Sample ID	Characteristics	Vegetation Phase of Plant	Date of Harvesting and Oil Distillation	Oil Yield (%)
1	Purple 2-15	127	Selected from the seed generation cultivar, which has certain increased decorative qualities due to the rich purple color of flowers	Full flowering	12.07.2016	1.15
2	Victoria 701-1	139	Flowers of a light blue colour and thin peduncles. Small number of flowers in inflorescences. This cultivar is late flowering and characterised by a large fraction of essential oil depending on weather conditions of the year from 1.2% to 1.6% of the fresh mass	Full flowering	25.07.2016	1.20
		312		Full flowering	25.06.2018	1.50
3	1-4-09	142	Selected from the seed generation, which is characterized by a compact habit of its bushes, blue flowers, and pleasant aroma	Full flowering	26.07.2016	0.90
4	Lidia	209	Characterized by thick peduncles with large inflorescences and light purple flowers	Full flowering	26.06.2017	0.90
		317		Full flowering	29.06.2018	0.85
5	1-3-16	210	Compact habit of bushes, blue flowers, and short dense inflorescences. It is hardy	Full flowering	03.07.2017	1.07
		311		The end of full flowering	23.06.2018	1.00
6	1-2-16	211	Semi-spreading habit of bushes, blue flowers, and large dense inflorescences. It is hardy	Full flowering	03.07.2017	1.20
		310		The end of full flowering	23.06.2018	0.83
7	Alba	215	Semi-spreading plants with semi-dense inflorescences and white flowers. It is hardy. The mass fraction of essential oil is small and ranges from 0.5% to 0.8% of the fresh mass	Full flowering	17.07.2017	0.65
		313		The end of full flowering	26.06.2018	0.65
8	Purple 2-1-17	220	Spreading habit of bushes with elongated peduncles and rich purple flowers. It has a pleasant aroma	Full flowering	19.07.2017	1.33
9	2-2-3	223	Small habit of bushes, narrow leaves, thin peduncles, and deep purple flowers. Early flowering	Full flowering	19.07.2017	0.65
10	2-4-6	224	Large habit of bushes and purple flowers. It is early flowering	Full flowering	19.07.2017	1.40
11	701-2	225	Small habit of bushes, light blue flowers, and thin peduncles. It has a large mass fraction of essential oil (ca 1.6% of the fresh mass)	Full flowering	19.07.2017	1.65
12	21/19	228	Large bush habit, thick peduncles, and light blue colour of flowers. It is hardy. Its plants have a pleasant aroma and a satisfactory amount of essential oil (1.0–1.1% of the fresh mass)	Full flowering	20.07.2017	1.00
13	Rosea	221	Small semi-spreading habit of bushes, thin peduncles with short head inflorescences, and pink colour of flowers. The mass fraction of essential oil is small and ranges from 0.45% to 0.60% of the fresh mass	Full flowering	19.07.2017	0.45
		308		The end of full flowering	22.06.2018	0.85

## Data Availability

The study did not report any data.

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
