# Peer review of "Chemical Composition of the Essential Oil of the New Cultivars of Lavandula angustifolia Mill. Bred in Ukraine"

_molecules, 2021, doi:10.3390/molecules26185681_

Round 1
Reviewer 1 Report
1) insert complete taxonomic names of the plant species throughout the text, paper title encompassed (it is Lavandula angustifolia Mill.). Check if names used are all accepted by using The Plant list or other databases.
2) line 34: The Family Lamiaceae
3) line 42: use Lavandula × intermedia Emeric ex Loisel. Check it throughout the text
4) line 48: use italics for L. angustifolia
5) When the name of a species is positioned at the start of a sentence or a paragraph, is right to write it in the full version; in the middle of a sentence, the second time you mention a species or in the case you use related entities, you can abbreviate the genus with a dot. For example, line 38-39: “Lavandula angustifolia,
commonly known as English Lavender (formerly known as L. vera or L. officinalis or true lavender)…” and so on. Check it throughout the text.
6) Minor English language check is required (plurals, word repetitions, etc.). Sometimes verbs are missing (i.e. sentence in line 178 -179)
7) statistical analyses should be better mentioned and described throughout the result and discussion section, and also in the table captions.
8) The file charged in the submission section shows a cut heatmap (the one in table 3), so it is impossible to check it accurately. The same is for table 5 and 6.
9) it is necessary to consider the vegetation phases when performing comparative analyses.
11) a paper like this could be of interest for the readers of Molecules. The article is written quite well and is surely catchy. However, the sampling methods cannot support the comparison between cultivars. Since in the paper an appreciable number of cultivars has been studied, I would suggest splitting the cultivars sampled at the end of the flowering stage from the rest, and perform comparisons between the other cultivars. According to several studies the essential oil composition undergoes changes during different flowering stages, so you cannot be sure that the composition that you observed were actually due to the variation in the cultivar itself and between years or to other physiological responses.
Daghbouche et al. 2020 Effect of phenological stages on essential oil composition of Cytisus triflorus L’Her
Hala I. Al-Jaber, Mahmoud A. Al-Qudah, Lina M. Barhoumi, Ismail F. Abaza & Fatma U. Afifi (2012) Variation in the essential oil composition of Eremostachys laciniata from Jordan at different flowering stages, Journal of Essential Oil Research, 24:3, 289-297
Reviewer 2 Report
The article has a well-defined structure, an introduction according to the development of the work, clearly describing the objectives it pursues. Likewise, the methodology followed to obtain each sample, as well as the characteristics of each cultivar. Although the study of the genus Lavandula is coarse, the article provides certain differences in the content of volatile compounds, probably due to climatic and genotypic issues.
On the other hand, the authors mention that their abundance percentages are semi-quantitative, so they cannot be used to evaluate the compliance required by the Ph. Eur. Or the ISO. However, the authors repeatedly make comparisons of their results with those required by said organizations, and use them as a reference. Following the initial mention, they contradict each other. I suggest making comparisons with works that have similarly obtained results, in order to establish a uniform judgment. Probably, that same factor is the cause of why not some of your results do not agree with the previously mentioned standards.
For this reason, it would be more appropriate to conduct its justification by contrasting it with studies in which the composition of the essential oils of Lavandula angustifolia is analyzed in climates similar to those of the cultivars analyzed, to check if the climate or the geographical region is a factor for the observed differences.
Tabla 3 faltan cultivares
In line 48 L. angustifolia is not in italics.
On line 204, the word "important" is repeated.
Tables 3, 5 and 6 are not complete, they do not seem to fit in the document. Columns for cultivars 8 - 13 are missing.
Author Response
please see the atachment

Reviewer 3 Report
Dear authors,
The information include in the paper can be of interest, but the paper has some problems that need to be addressed before even considered for publication. Please response the comments point-by point and see the attached file.
Sincerely Yours,
Journals Referee

Reviewer 4 Report
Comments:
The manuscript reports the “Chemical composition of the essential oil of Lavandula angustifolia of Ukrainian origin”. The manuscript should be improved, especially the English language. Grammatical errors were found throughout the manuscript. Linguistic certificate after editing should be provided.
Title:
The title should revise.
Abstract:
The abstract is lengthy and not well-structured. Please reconstruct to make it more organised and objective. Authors should give a brief background of the plant in the first sentence.
Line 15: Avoid to use “about”. Be accurate to state how many components were identified.
Introduction:
In general, some statements are not linked to each other, such as: Paragraph 1 and Paragraph 2 as well as Paragragh 2 and Paragragh 3. Authors are recommended to re-structure the Introduction and consolidate the content.
Line 35: Lavandula --> italic [genus and species names are in italic]
Line 37: Repeating statement as Line 34.
Line 38: Authors should use “L. angustifolia” instead of “Lavandula angustifolia” after first introduced the genus. Please revise throughout the manuscript.
Line 45: Lavandula angustifolia --> L. angustifolia [and so on]
Line 48: L. angustifolia --> italic
Line 50: The “True lavender oil” is referring to Lavandula angustifolia? Authors should define true lavender oil before elaborating. Did the true lavender oil relate to present study?
Line 61: Lavandula angustifolia --> L. angustifolia [and so on]
Line 97: moderate amount
Line 101: L. angustifolia [and so on]
Line 119: Figure 1 is not in high resolution. Resend.
Line 121-123: Reconstruct the sentence.
Results and Discussion:
It is recommended to combine Table 1 and Table 2 into one so that all components from different cultivars could be visualised at once. Cite the reference for literature LRI in Tables.
Line 181: “It seems that the terpinen-4-ol content was higher than values usually reported in the literature”. Please provide literature value of terpinen-4-ol to justify the claim.
Line 188: (0,2-0,4%) --> Please correct.
Line 191: State the requirement of the Ph. Eur. for the lavandulol content.
Line 193: Revise Figure 3 so that it showed only 10 components, aligned with Table 3.
Line 194: Table 3 is not showing completely. Revise the caption.
Line 195: Be consistent in using “component” throughout the manuscript.
Line 200: manufacturers
Line 207: Be consistent in using abbreviation throughout the manuscript, European Pharmacopeia and etc. It has been used in abbreviated form in Line 191.
Line 210: L. angustifolia
Line 212: Be consistent in using 10th or X edition.
Line 213: L. angustifolia
Line 213-216: It should not have repeated what have been shown in Table 4 in the text.
Line 216: Be consistent in using abbreviation, once introduced, please use its abbreviated form throughout the manuscript.
Line 221-222: Reconstruct the sentence.
Line 229: “… which are different from zero”. Could authors elaborate it?
Line 269 & Line 271: Table 5 & 6 are not showing completely in the manuscript.
Table 6: β-ocimene (E) --> Please either use (E)-β-ocimene or trans-β-ocimene. Be consistent.
Table 6: β-ocimene (E) --> 0,5-6 --> 0.5?
Line 273-275: Reconstruct the sentence.
Line 300-301: Abbreviation NIR is not introduced previously.
Line 303: How did authors classify components as major, main or minor? Based on what justifications?
Line 360-361: Please provide the reference regarding the claim.
Line 362: Different years but same month?
Line 370: Please avoid using “we” in the manuscript. Please check throughout the manuscript.
Line 377-379: Reconstruct the sentence.
Figure 5: Authors should include all photos for all cultivars (1-13).
Line 412: What does it mean by selection process?
Line 407-427: It is recommended to group all cultivars which are used for same purposes together rather than repeating one by one.
Overall, it is recommended to tabulate the different composition of major components present in lavender oils from different origins, extraction methods, plant maturity stages and other factors.
As stated by authors, the present study has some drawbacks. The relative % of each component is not calculated based on GC-FID. Thus, the results are not suitable to use in comparison to the standards set by the authorised bodies. Authors should avoid discuss lengthy regarding this.
Authors stressed that 13 lavender cultivars are new but no preliminary genetic/DNA information was provided to justify the claim.
Materials and Methods:
Line 442: Authors should include the “month” in which the plant materials were collected.
Authors should provide the weight of the plant materials used in each extraction.
Are the plants’ cultivars authenticated by a botanist?
Line 457-458: “Each chromatographic analysis was repeated three times”. Is the chromatographic repetition from the same extraction or different extraction of the same cultivar?
Conclusion:
Line 484: Authors should discuss the environmental factors that might affect the chemical composition in the Results and Discussion before making this conclusion. The same for weather conditions. The conclusion should conclude the findings found in the study.
References:
Inconsistencies in reference style were noticed, please format according to Journal’s requirement.
Round 2
Reviewer 1 Report
Dear authors, In the proof round, check italics (for species), commas, plurals etc .Reviewer 4 Report
Comments:
Line 34: L. angustifolia --> italic
Line 95: publication --> literature
Line 134: * tentative identification --> put as footnote below the Table
Table 1:
- Make the numbers 2, 4, 5, 6 which represent the cultivars as centered
(The same for 7 and 13 in Table 2)
- Components 17-18: p and o are in italic (cis, trans, α, β and etc. are all in italic) Please revise throughout the manuscript
Line 215: cultivars
Line 218: cultivars
Line 225: ninethirteen?
Line 396: One-hundred grams of flowering parts of L. angustifolia were used for each distillation.